# Increasing Step Rate Affects Rearfoot Kinematics and Ground Reaction Forces during Running

**DOI:** 10.3390/biology11010008

**Published:** 2021-12-21

**Authors:** Kathryn A. Farina, Michael E. Hahn

**Affiliations:** Bowerman Sports Science Center, Department of Human Physiology, University of Oregon, Eugene, OR 97403, USA; kfarina@uoregon.edu

**Keywords:** step rate, gait retraining, cadence, injury, pronation, tibial rotation

## Abstract

**Simple Summary:**

Excessive movements, or inadequate timing in movement patterns, during running may contribute to the development of some running-related injuries. Specifically, excessive movement at the rearfoot, influencing lower leg rotation, has been a focus on different running-related injuries. One method to change how the lower limbs move is to increase step rate, or cadence. There is little research available describing how the rearfoot is affected by changes in step rate; therefore, the primary purpose of this study was to evaluate the effects of increasing step rate on rearfoot motion during running. Reflective markers were placed on twenty runners’ lower legs and feet in order to capture leg and foot movements while running on a treadmill at the runners’ preferred speed and step rate. Step rate was increased by 5% and 10%, while runners were cued by a metronome. Three-dimensional rearfoot motion was calculated during the stance phase (foot in contact with the ground) of running. The main finding of this study was that increasing step rate decreased peak rearfoot and lower leg rotation. These findings may be useful for rehabilitation for some running-related injuries.

**Abstract:**

Relatively high frontal and transverse plane motion in the lower limbs during running have been thought to play a role in the development of some running-related injuries (RRIs). Increasing step rate has been shown to significantly alter lower limb kinematics and kinetics during running. The purpose of this study was to evaluate the effects of increasing step rate on rearfoot kinematics, and to confirm how ground reaction forces (GRFs) are adjusted with increased step rate. Twenty runners ran on a force instrumented treadmill while marker position data were collected under three conditions. Participants ran at their preferred pace and step rate, then +5% and +10% of their preferred step rate while being cued by a metronome for three minutes each. Sagittal and frontal plane angles for the rearfoot segment, tibial rotation, and GRFs were calculated during the stance phase of running. Significant decreases were observed in sagittal and frontal plane rearfoot angles, tibial rotation, vertical GRF, and anteroposterior GRF with increased step rate compared with the preferred step rate. Increasing step rate significantly decreased peak sagittal and frontal plane rearfoot and tibial rotation angles. These findings may have implications for some RRIs and gait retraining.

## 1. Introduction

Greater frontal and transverse plane motion in the lower limbs during running have been thought to play a role in the development of some running-related injuries (RRIs) [1,2,3]. Increased tibial rotation during the stance phase of running has been targeted as a possible mechanism contributing to patellofemoral pain syndrome and iliotibial band syndrome in runners by causing increased joint compression force on the patella, and friction over the iliotibial band insertion [4,5,6]. Tibial rotation has been shown to be coupled to motion at the rearfoot, because these segments are linked through the subtalar and talocrural axes [7,8]. Thus, motion at the rearfoot may have an influence on the amount of segmental motion propagating up the kinetic chain [7]. In normal running locomotion, a slightly supinated foot makes contact with the ground and the calcaneus everts as the subtalar joint begins to pronate, causing the talus to move medially and adduct, thus internally rotating the tibia due to the tight articulations between the subtalar, talocrural, and tibiotalar joints [1]. Similarly, once the foot moves into midstance, the foot begins to supinate and the knee extends to prepare for the foot leaving the ground, causing the tibia to externally rotate [1].

When a mismatch arises in the timing of these coupling events, or one segment displays excessive motion, rotation of the tibia may conflict with rearfoot eversion, and cause increased stresses to be placed on other soft tissues or bones [9]. Indeed, one of the theories for the development of Achilles tendinopathy centers around this conflicting motion between the start of tibial external rotation with knee extension and prolonged rearfoot eversion, causing increased stress on the Achilles tendon [9]. Although a causal relationship between increased rearfoot motion and the development of running injuries has not been firmly established, there is conflicting evidence suggestive of some type of link. Runners with Achilles tendinopathy have been reported to have increased rearfoot eversion, or duration of eversion, compared with healthy controls [10,11]. Similarly, excessive rearfoot eversion has been reported in runners with tibial stress fractures [12], medial tibial stress syndrome [10], and patellofemoral pain syndrome [13]. 

Attention has been given to rearfoot eversion in evaluating RRIs, and it has often been used as a surrogate measure to describe pronation, which has long been thought to play a role in RRI development [1,9]. However, pronation is a complex motion, involving movement between the forefoot, rearfoot, and ankle, subsequently influencing tibial rotation [7]. Indeed, movement at the subtalar joint involves motion in multiple planes, and has been shown to be linked to tibial rotation in both the transverse and frontal planes [8]. Despite the link between rearfoot and tibial rotation, and the possible effects on RRI development, little is known about methods for altering frontal and transverse plane rearfoot motion, if such an alteration influences injury risk, or whether such methods could be used for rehabilitation after injury. 

For many RRIs that appear to arise from injurious segmental motion patterns, various methods of gait retraining have been employed to alter the movement pattern of interest to decrease pain and return the runner to full training. Promising results have been shown when increased hip adduction has been targeted in an effort to rehabilitate runners with patellofemoral pain and iliotibial band syndrome [14,15,16]. Runners completing a gait retraining program aimed at decreasing excessive hip adduction were able to reduce pain levels and return to pre-injury training volumes [14,15,16]. These previous investigations have used real-time visual feedback displaying pelvic angles or used a mirror to focus the runners’ attention on hip movement. Although successful, these methods generally require the runner to visit a lab or clinic for many gait retraining sessions. One simple, effective, and low-cost gait retraining method requiring minimal supervision is to have runners increase their step rate. After foot strike, the ground reaction force propagates through the subtalar joint, contributing to rearfoot eversion, a necessary function enabling foot pronation in order to aid in shock absorption, and help the foot form a rigid lever to prepare for push-off [7]. Increasing the step rate has been shown to decrease peak ground reaction forces and loading rates, which may require less energy absorption from the lower extremity musculature and joints [17,18,19,20,21]. Increasing step rate effectively draws the foot closer to the body center of mass at ground contact, reducing vertical oscillation of the center of mass, thereby reducing the energy absorbed by the lower limbs, and altering joint kinematics [20,21]. For example, increasing step rate has been shown to decrease peak hip adduction [20,22,23,24], peak knee abduction [25], and peak rearfoot eversion [2], all of which have been implicated in the development of specific RRIs [10,15,26]. 

Few studies have looked at how non-sagittal plane rearfoot motion is affected by an increasing step rate. With the linkage between rearfoot motion and tibial rotation, and the potential development of RRIs, it is important to discover how these motions can be modified. Therefore, the purpose of this study was to evaluate the effects of increasing step rate on kinematics at the rearfoot in the sagittal, frontal, and transverse planes during the stance phase of running. A secondary purpose was to confirm how ground reaction forces throughout the stance phase are adjusted with increased step rates. We hypothesized that there would be significant reductions throughout stance in rearfoot angles and ground reaction forces. 

## 2. Materials and Methods

Twenty runners (nine female) were recruited for participation in this study (Table 1). Participants had to be between the ages 18–65, running at least 15 miles per week, and be running pain-free at the time of data collection. Prior to data collection, participants provided written informed consent, approved by the University of Oregon Institutional Review Board.

Forty-three retroreflective markers were placed bilaterally on the lower limbs and pelvis to define pelvis, thigh, shank, rearfoot, and forefoot segments. Standardized, neutral running shoes (Brooks Launch) were used by each of the participants, in which windows were cut to place markers directly on the foot [27]. Participants performed a static trial, after which markers on the medial and lateral malleoli, femoral epicondyles, and greater trochanters were removed so as not to interfere with running motion. 

Three-dimensional marker trajectories were collected using an 8-camera motion capture system (Motion Analysis Corp., Rohnert Park, CA, USA), and ground reaction force data were collected using an instrumented treadmill (Bertec, Columbus, OH, USA) at 200 and 1000 Hz, respectively. Participants performed three running trials on the treadmill consisting of three minutes each. Kinematic and kinetic data were recorded for twenty strides during the final minute of each trial. The first trial consisted of the participants running at a self-selected ‘easy’ pace to determine their preferred running step rate. The step rate was assessed by counting the number of foot falls during a 20 s period, then multiplying this number by 3, to determine number of steps per minute. Calculations of +5% and +10% increases over the preferred step rate were then determined for each subject. In the following two trials, a metronome was set to the calculated increase in step rate, and participants were instructed to match their foot falls to the beat of the metronome. The +5% trial was followed by the +10% trial. Subjects were given the first two and a half minutes of each trial to acclimate to the increased step rate. Step rates were calculated during the final minute of each trial, once before and once after data were recorded, to ensure that participants had modified their step rate.

### 2.1. Data Analysis

A customized post-process script for traditional gait analysis was written in MATLAB (The MathWorks, Natick, MA, USA) to calculate rearfoot kinematics from the right foot throughout the stance phase. The stance phase was defined as when the vertical ground reaction force exceeded 5% of the participant’s body weight [28]. This threshold defined the start and end of each foot contact with the ground. The rearfoot coordinate system origin was defined as the midpoint between the medial and lateral calcaneus markers, with the *x* axis pointed laterally, *y* axis pointed anteriorly, and *z* axis directed superiorly. Raw marker coordinate and force platform data were dual-pass-filtered using a 4th order lowpass Butterworth filter with a 20 Hz cutoff frequency. Joint angles were calculated using a Cardan sequence of flexion/extension, inversion/eversion, adduction/abduction. It has previously been noted that sagittal plane motion between the tibia and calcaneus primarily occurs through the talocrural joint, whereas frontal and transverse plane motions occur at the subtalar joint [29]. In this study, rearfoot sagittal plane angle primarily reflected talocrural dorsiflexion/plantarflexion, with inversion/eversion and adduction/abduction reflecting subtalar joint motion [29]. Rearfoot angles were calculated with respect to the shank segment, with the exception of the transverse plane angle calculated as the shank segment with respect to the rearfoot segment to reflect tibial rotation [29,30]. 

Foot strike index (FSI) values at initial contact were calculated in order to assess changes in foot strike pattern with the increasing step rate. Center-of-pressure (COP) data were first transformed into the rearfoot coordinate system. Then, the FSI was calculated as the longitudinal difference (heel to toe) between the COP and the heel marker at initial contact with the ground. This difference was then divided by the total foot length to obtain a percentage of foot length [31]. 

Twenty right-foot strikes, defined by the stance phase threshold previously described, were identified. Rearfoot angles and ground reaction forces were calculated for each stance phase, averaged across twenty right-foot strikes, and normalized to 101 data points. Peak values for rearfoot dorsiflexion, eversion, and tibial internal rotation, and vertical, braking, and propulsive ground reaction forces were extracted by finding the peak value within each curve. 

### 2.2. Statistical Analysis

A repeated-measures analysis of variance (ANOVA) (α = 0.05) was used to test the effects of the three step rate conditions on the discrete peak values listed above, and FSI at initial contact. In cases of a significant main effect, a Bonferroni correction was used in the post hoc analysis to determine significant differences between conditions. Statistical analyses were performed in SPSS v27 (IBM SPSS Statistics, Chicago, IL, USA). 

Statistical parametric mapping (SPM) was used to analyze differences in the normalized stance phase time series curves between conditions for the rearfoot angles and ground reaction forces. The open-source “SPM1D” package was used in MATLAB [32]. In this approach, a repeated-measures ANOVA over the normalized time series was used to determine any significant differences between the three conditions. If statistical significance was reached, post hoc *t*-tests over the normalized time series were used to determine significant differences between conditions. These analyses involve computing a test statistic at each time point in the curve, calculating a critical threshold at which only the α % (5%) of smooth random curves would be expected to cross, and finally, calculating the probability that specific points, or clusters of points, could have exceeded the critical threshold due to a random field process. The final analysis effectively produces suprathreshold clusters, or areas of the time-normalized curve, which are significantly different from each other (if any significant differences are found) [32,33]. 

## 3. Results

### 3.1. Peak Variables Analysis

Peak rearfoot angles were significantly different between step rate conditions in all planes of motion (Table 2). Increasing step rate significantly decreased the peak dorsiflexion angle at the +5% (*p* = 0.010) and +10% step (*p* = 0.001) rate conditions compared with the preferred condition. The +10% condition showed a significantly decreased peak dorsiflexion angle compared with the +5% condition (*p* = 0.016). Peak eversion was significantly decreased from the preferred step rate condition in the +5% (*p* = 0.013) and the +10% condition (*p* = 0.008). Peak tibial internal rotation was significantly decreased in the +10% condition compared with the preferred condition (*p* = 0.037).

The peak vertical ground reaction force was significantly decreased in the +10% step rate condition as compared with the preferred (*p* = 0.005) and +5% (*p* = 0.010) conditions. Peak braking force was significantly reduced in the +5% condition compared with the preferred condition (*p* = 0.010), and in the +10% condition compared with the preferred (*p* = 0.002) and +5% conditions (*p* = 0.004). There were no significant differences found for FSI at the initial contact between conditions (Table 2). 

### 3.2. Time Series Analysis

The preferred condition displayed significantly increased sagittal plane rearfoot angle between 45.1% and 59.2% of stance (*p* = 0.018) compared with the +10% condition. Compared with the +5% condition, the sagittal plane rearfoot angle in the +10% condition was significantly reduced between 35.3% and 51.2% (*p* = 0.022) and between 90.2% and 100% of stance (*p* = 0.037) (Figure 1). 

The frontal plane rearfoot angle was significantly reduced in the +5% condition between 30.8% and 42.1% of stance (*p* = 0.031) and in the +10% condition between 20.4% and 44.0% of stance (*p* = 0.0060) as compared with the preferred condition (Figure 2). 

The transverse plane tibial rotation angle was significantly reduced in the +10% condition between 2.6% and 34.9% of stance (*p* < 0.001) compared with the preferred condition, and significantly reduced in the +10% condition between 6.1% and 31.4% of stance (*p* = 0.0059) compared with the +5% condition (Figure 3). 

In the +10% step rate condition vertical ground reaction force was significantly reduced compared with the preferred condition between 6.7–12.0% (*p* = 0.023) and between 38.8% and 51.6% (*p* < 0.001) of stance, and compared with the +5% condition between 2.6% and 13.3% (*p* = 0.0033) and between 35.0% and 47.8% (*p* < 0.001) of stance. In the anteroposterior direction, the preferred condition displayed significantly greater braking force compared with the +5% and +10% conditions between 21.2% and 26.3% (*p* = 0.032) and 14.6% and 29.9% (*p* < 0.001), respectively. The +10% condition also displayed significantly decreased braking force compared with the +5% condition between 15.5% and 29.4% (*p* < 0.001) of stance. There were no significant differences across the stance phase found between conditions in the mediolateral ground reaction force.

## 4. Discussion

The purpose of this study was to analyze motion of the rearfoot during the stance phase of running with an increased step rate. Supporting our hypothesis, decreases in peak rearfoot angles were observed in the sagittal, frontal, and transverse planes. Similar to previous investigations, peak rearfoot dorsiflexion was decreased with an increasing step rate. A +10% increase in step rate was reported to decrease ankle dorsiflexion at midstance by around 8%, corresponding to a 2.5° decrease in ankle dorsiflexion at midstance [34]. Likewise, other investigations implementing a +10% increase in step rate reported decreases in peak dorsiflexion angle of approximately 2°–2.5°, compared with the preferred step rate [25,35]. These investigations observed a slightly greater decrease (1°) in ankle dorsiflexion than the results observed in the present study. This discrepancy may be due to differences in the average preferred running speeds and step rates, both of which were slightly greater in the present study. These previous investigations have also evaluated ankle dorsiflexion, whereas the present study observed rearfoot angles, with an origin in the calcaneus segment. This difference may have influenced the observed differences between the present study and prior work. However, as noted previously, the majority of rearfoot plantar dorsiflexion occurs at the talocrural joint [29], indicating that there would be little functional difference between these calculated angles.

In the frontal plane, peak rearfoot eversion was decreased in the +5% and +10% conditions compared with the preferred condition. Boyer and Derrick found a significant linear trend for a decrease in peak ankle eversion as stride length was decreased (thereby increasing step rate, assuming a constant speed), and observed a 0.6° decrease in peak rearfoot eversion as stride length decreased [2]. We observed a slightly greater average decrease in peak rearfoot eversion angle from the preferred rate to +10% condition: 0.79°. Again, this difference may be due to differences in joint measurement definitions. Another investigation comparing a +10% increase and preferred step rate reported only a 0.24° decrease from the preferred to increased step rate condition, which was not significant [36]. Although these authors also evaluated rearfoot motion with a coordinate system similar to the present study, participants’ preferred step rate was lower than in the present study, and these authors utilized an overground running protocol, differing from our use of the treadmill, which may have had an effect on joint kinematics [37]. 

Few studies have evaluated the effects of increasing step rate on tibial rotation. Boyer and Derrick found the peak knee internal rotation angle to decrease by 0.4° from a preferred to 10% decreased stride length [2]. The present study observed peak tibial internal rotation to decrease, on average, by 0.8° from the preferred to +10% step rate condition. More similar to the results observed in the present study, peak knee rotation during stance was found to decrease by approximately 0.5° with a 10% increase in step rate [25]. Although not directly manipulating step rate, Pohl and Buckley found the peak tibial internal rotation angle to decrease by almost 3° between a rearfoot-strike and toe-strike pattern [38]. It has been suggested that running with an increased step rate may be associated with transitioning from a rearfoot- to a forefoot-strike pattern [17]. We did not observe a significant difference in FSI between step rate conditions in the present study, but there was a trend for subjects to land more anteriorly on the foot, closer to a mid-foot strike pattern. This may indicate that the subjects were beginning to demonstrate early patterns of transitioning from a rearfoot-strike to forefoot-strike pattern. 

Analysis using SPM allows for differences between conditions to be viewed throughout the stance phase, as opposed to only at discrete time points. This approach provides a more complete view of when changes occur throughout the stance phase. There were significant differences throughout the stance phase for rearfoot angles in the sagittal and frontal planes. In the sagittal plane, differences occurred close to peak dorsiflexion at toe-off. Similarly, the frontal plane rearfoot angle was significantly reduced near the period of peak eversion in both the +5% and +10% conditions, as compared with the preferred condition. Tibial rotation angle displayed a greater range of significant differences during stance, with approximately the first 30% of stance being significantly reduced in the +10% condition compared with the preferred and +5% conditions. Interestingly, differences in rearfoot eversion were not observed until the latter part of this early stance period, suggesting that there may be other mechanisms contributing to the differences in tibial rotation in the first 30% of stance. Some authors have suggested that tibial internal rotation could also be caused by more proximal mechanisms, such as changes at the hip [39]. Increasing step rate has been shown to affect motion at the hip [2,20,23,25], adding support for hip compensation to impact these changes seen in tibial rotation early in stance. This period of significance in the first 30% of stance did not include peak tibial internal rotation; however, significant differences were observed when the discrete values were compared between conditions. Peak eversion occurred approximately 10% earlier in the stance phase than peak tibial internal rotation. The significant cluster in the frontal plane angle also occurred about 10% earlier in the stance phase than peak tibial internal rotation, suggesting that the changes in rearfoot eversion could have influenced peak tibial internal rotation.

The secondary purpose of this study was to assess changes in ground reaction forces with increased step rate. Peak vertical ground reaction force was significantly reduced in the +10% condition compared with the preferred and +5% conditions. Previous investigations have also reported decreases in peak vertical ground reaction force of 2.6% and 3.5% with a 10% increase in step rate [34,40]. Similarly, a +10% increase in step rate was shown to significantly decrease the peak vertical ground reaction force by 0.6 N/kg from the preferred condition [20]. The results from the present study strongly agree with these previous findings, because we observed an average 0.07 BW decrease from the preferred to +10% condition, corresponding to approximately a 2.8% decrease between conditions. We also observed a significant reduction in peak braking force in the +10% condition compared with the preferred and +5% condition, and the +5% condition compared with the preferred condition. These decreases corresponded to a 5.7% decrease from the preferred to +5% condition, and an additional 3% decrease from the +5% to +10% condition. Additionally, the +10% condition displayed an 8.6% decrease compared with the preferred condition. Lenhart et al. observed a 5.5% decrease in the anterior–posterior ground reaction force maximum with a 10% increase in step rate [34]. Other reports have also detected an approximately 9% decrease in peak anterior–posterior ground reaction force with a 10% decrease in stride length [41], correlating well with the results in the present study.

Significant differences in the vertical ground reaction force were observed between 38% and 51% of stance when comparing the preferred and +10% condition. The +5% condition also displayed significantly decreased vertical ground reaction force between 35% and 47% of stance. These results are likely related to the significant differences observed in the peak ground reaction forces, because these time periods correspond closely to the timing of the peak vertical ground reaction force. There was also a small time period early in stance, between 2% and 13%, found to be significantly different between conditions, possibly indicating differences in the rate of loading in the vertical ground reaction force. In the anterior–posterior direction, similar results were observed. A significantly decreased braking force was observed with increasing step rate in the period between 14% and 30% of stance, likely also reflecting both rate of loading and peak braking force. 

Both foot pronation and knee flexion are essential components of gait to enable efficient shock absorption upon ground contact when the foot moves through the stance phase during running [42]. As the subtalar joint everts and the talus adducts, the tibia is forced to internally rotate [7]. Then, as the knee starts to extend after the foot has reached midstance, the tibia begins to externally rotate, just as the foot moves into supination to prepare for toe-off [7]. It has been thought that if the timing is poor between the transitions to pronation/supination and tibial internal/external rotation, the distal and proximal ends of the tibia will experience conflicting rotations from the talus and knee [9,42]. These conflicting rotations could then lead to the development of RRIs [9]. There appears to be a coupling between tibial rotation and rearfoot motion [8]; however, it remains unknown if altering these movements has an impact on RRI development. There is some evidence to suggest that adjustments to step rate can influence forces and stresses in the lower extremities associated with common RRIs. Increasing step rate has been shown to decrease peak knee flexion during stance, contributing to a reduction in patellofemoral joint loading [34]. Similarly, tibiofemoral joint forces have been shown to decrease with a 10% decrease in step length [22]. Achilles tendon stress has also been shown to decrease with a 5% increase in step rate [43]. The present study supports these past findings: we found significant reductions in rearfoot motion, tibial rotation, and vertical and braking ground reaction forces with an increased step rate. Despite these findings, further research is needed to solidify the connections between changes in rearfoot motion, tibial rotation, and RRI development.

The results from this study indicate that increasing step rate has the potential to alter rearfoot and tibial motion, as well as ground reaction forces. The observed decreases in ground reaction forces in the present study may diminish the need for shock absorption, thus leading to a reduction in subtalar joint motion. As motion at the subtalar joint is transferred to the tibia [42], reductions in subtalar joint motion could also lead to decreased tibial rotation. Increasing step rate may allow for proximal changes which influence tibial rotation as well. Increasing step rate has been shown to decrease both ground reaction forces and knee flexion during stance [34]. With the reduced need for shock absorption, the knee may not be forced into as much flexion. This could alter the timing between the changes in knee flexion/extension, because the knee would not have to move through as great of a range of motion from reduced peak flexion. Improved timing between the shift to external tibial rotation when the knee begins to extend could allow for a reduction in possible conflicting rotations between the proximal and distal ends of the tibia. 

A limitation of this study was not controlling for whether runners were habitual rearfoot- or forefoot-strike runners. Differences in ankle and rearfoot kinematics have been observed between rearfoot- and forefoot-strike runners in previous investigations [38]. However, most runners (n = 16) in this sample tended to utilize a rearfoot strike pattern, and our participants did not significantly alter their foot strike pattern with the increased step rate. When comparing the rearfoot- and forefoot-strike runners, no differences were observed in the variables of interest. The average step rate of the runners in this study was relatively high before the increased step rate conditions. Although this study was focused on how increased step rate influenced rearfoot kinematics, increasing step rate may have a ceiling effect [19], and it may not be feasible for the average runner to increase their step rates to the level observed in some of the subjects’ +10% trials. Screening for runners with lower step rates would increase the applicability of this study because runners with lower step rates would likely find more benefit from increasing their step rate and may experience more pronounced effects from an intervention. This study also only tested the acute effects of increased step rate on measures of rearfoot kinematics and ground reaction forces. Previous investigations implementing at-home or in-lab gait retraining sessions over the course of multiple weeks or months have shown that participants are able to effectively alter their step rate [19,22,23,44,45,46,47]. However, the longitudinal effects on rearfoot motion, and whether the modifications in rearfoot angles and ground reaction forces would still be observed after implementing a longer-term gait retraining protocol, remain unknown. Finally, we had participants complete the running protocol on a treadmill. Results from this study may not be generalizable to overground running, because there have been differences observed between treadmill and overground running [48]. Future work could look to analyze rearfoot motion in overground running outside of the laboratory by using portable technologies capable of motion capture data collection [49].

## 5. Conclusions

The results from this study suggest that increasing step rate alters rearfoot motion in the sagittal and frontal planes, as well as tibial rotation. These outcomes have implications for many RRIs such as Achilles tendinopathy, patellofemoral pain, and tibial stress injuries, because reducing excessive rearfoot motion and tibial rotation may prove beneficial in rehabilitating or preventing some of these common RRIs. Further research is needed to quantify the effects these changes in rearfoot and tibial motion have on soft tissue and bone loads. 

## Figures and Tables

**Figure 1 biology-11-00008-f001:**
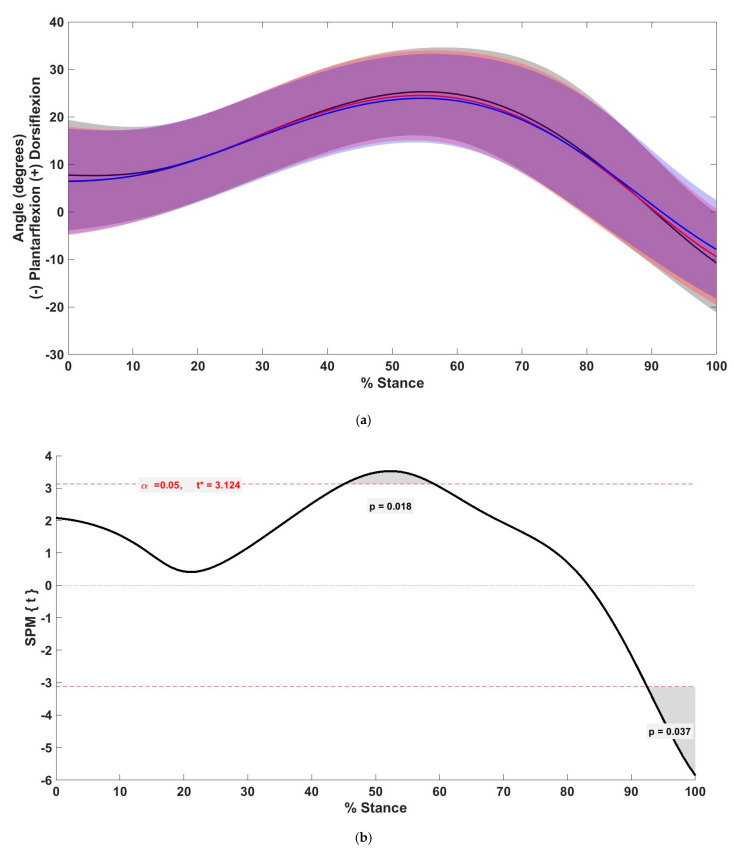
(**a**) Mean (solid line) and standard deviation (shaded) of the sagittal plane rearfoot angle between preferred (black), +5% (red), and +10% (blue) step rate conditions. (**b**) *t*-values of SPM post hoc comparison between preferred and +10% conditions for sagittal plane rearfoot angle. Dashed red lines indicate critical threshold (α = 0.05). Gray shaded area indicates regions with statistically significant differences between the preferred and +10% condition.

**Figure 2 biology-11-00008-f002:**
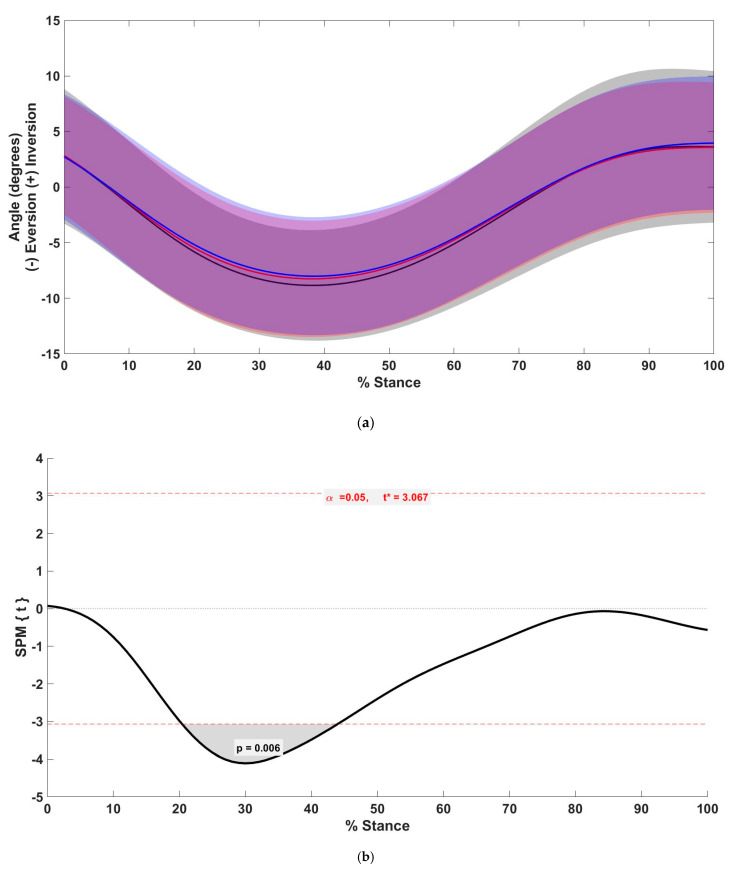
(**a**) Mean (solid line) and standard deviation (shaded) of the frontal plane rearfoot angle between preferred (black), +5% (red), and +10% (blue) step rate conditions. (**b**) *t*-values of SPM post hoc comparison between preferred and +10% conditions for frontal plane rearfoot angle. Dashed red lines indicate critical threshold (α = 0.05). Gray shaded area indicates regions with statistically significant differences between the preferred and +10% condition.

**Figure 3 biology-11-00008-f003:**
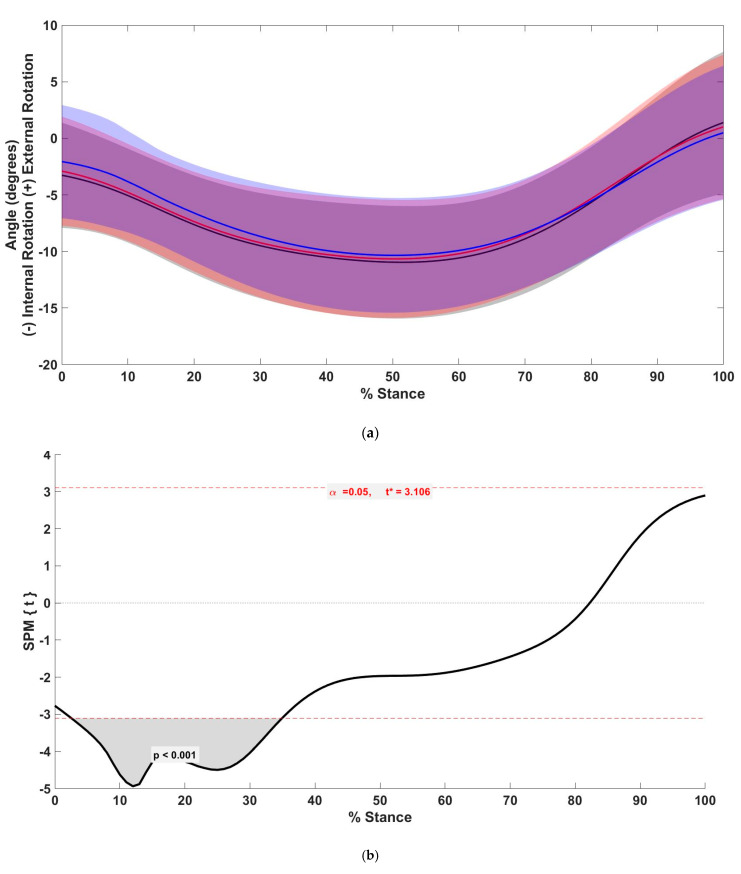
(**a**) Mean (solid line) and standard deviation (shaded) tibial rotation angle between preferred (black), +5% (red), and +10% (blue) step rate conditions. (**b**) *t*-values of SPM post hoc comparison between preferred and +10% conditions for tibial rotation angle. Dashed red lines indicate critical threshold (α = 0.05). Gray shaded area indicates regions with statistically significant differences between the preferred and +10% condition.

**Table 1 biology-11-00008-t001:** Subject characteristics, preferred running pace, and step rates (steps/min).

	n = 20
Age	24.9 ± 8.66
Height (cm)	173.69 ± 9.83
Mass (kg)	64.69 ± 11.27
Miles per week	34.50 ± 17.08
Preferred Pace (m/s)	3.33 ± 0.38
Preferred Step Rate	175 ± 7
+5% Step Rate	185 ± 9 *
+10% Step Rate	192 ± 9 *^#^

* denotes significant difference from preferred condition; ^#^ denotes significant difference from +5% condition (*p* < 0.05).

**Table 2 biology-11-00008-t002:** Peak rearfoot angles (degrees), peak tibial internal rotation (degrees), peak vertical, propulsion, and braking ground reaction forces (BW), and foot strike index (% of foot length) at initial contact for the preferred, +5%, and +10% step rate conditions.

	Preferred	+5%	+10%
Peak Dorsiflexion	25.85 ± 9.28	24.94 ± 9.60 *	24.27 ± 9.53 *^#^
Peak Eversion	8.97 ± 5.15	8.40 ± 5.44 *	8.18 ± 5.52 *
Peak Internal Rotation	11.39 ± 5.06	10.94 ± 5.37	10.58 ± 5.27 *
Peak Vertical	2.50 ± 0.22	2.47 ± 0.25	2.43 ± 0.24 *^#^
Peak Propulsion	0.30 ± 0.04	0.29 ± 0.04	0.28 ± 0.04
Peak Braking	0.35 ± 0.04	0.33 ± 0.04 *	0.32 ± 0.05 *^#^
Foot Strike Index	35.44 ± 13.75	37.63 ± 18.35	40.95 ± 19.93

* indicates significant difference from the preferred condition; ^#^ indicates significant difference from the +5% condition (*p* < 0.05).

## Data Availability

The data presented in this study will be made available upon reasonable request.

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
