# Peer review of "Increasing Step Rate Affects Rearfoot Kinematics and Ground Reaction Forces during Running"

_biology, 2021, doi:10.3390/biology11010008_

Round 1

Reviewer 1 Report

The research work presents an interasting study about the evaluation of rearfoot kinematics and ground reaction forces during running. 

The paper is well written and the structure is correct for a journal pubblication. 

In general, the paper present in clear way the whole study and the results are clear and complete.

The authors should add several works that have already exploits different motion capture systems for either sport activities or rehabiliation processes. For example:

  • DOI: 10.1007/978-3-030-51064-0_33
  • DOI: 10.3390/s20216273

The matlab software module should be better described:

  1. Has the gait cycle analysis developed specifically for this research work?
  2. Why did you choose the midpoint to compute the position of the rearfoot? Are there any reserach works in literature to confirm your choice? The authors should better justify this choice.
  3. An image about the main workflow of the data analysis should be added to better clarify the use of Matlab.

After the proposed modification, the paper can be considered ready for pubblication.

Author Response

We thank the reviewers for taking the time to review this manuscript, and for the positive feedback. We have revised the manuscript accordingly, and are confident that the manuscript has been improved through the review process. Please find point by point responses to comments below.

Reviewer 1

The authors should add several works that have already exploits different motion capture systems for either sport activities or rehabilitation processes. For example:

  • DOI: 10.1007/978-3-030-51064-0_33
  • DOI: 10.3390/s20216273

Response:

  • We are unsure of the relevance of these two publications in relation to our manuscript. The first referenced work, ‘Advances in Simulation and Digital Human Modeling,’ appears to contain works primarily related to AI, Human-Machine Interaction, IMUs, and new approaches to motion capture analysis. The ‘Motion Capture and Analysis’ section in this work does not appear to have any works relating to using traditional motion capture and running. We do not see the utility in referencing any of these works, as our methodology uses traditional motion capture technology during running and does not attempt to offer a new method of motion capture analysis or collection. Therefore, there is no relevant connection with the suggested work.
  • The second recommended publication, similarly, does not appear to be related to the current manuscript. The suggested work uses an alternative method to motion capture that is low-cost and easier for medical professionals to utilize for tracking spinal cord motion during rehabilitation. Although finding low-cost and easily accessible alternatives to traditional motion capture is important, we did not attempt this in the present study nor make claims to use our methodology as an alternative to traditional motion capture. Therefore, the recommended work does not appear to relate to the present study, as we are not proposing an alternative to traditional motion capture and did not use an alternative approach to motion capture. Nor did we present any data on spinal motion. Lastly, we analyzed running movement on a treadmill with techniques that are widespread in the field of running biomechanics. The nature of our protocol and measurement techniques and does not have a relation to the spinal cord patients in the suggested reference.

The matlab software module should be better described.

Response:

  • We have added additional descriptions of the steps taken for calculating our variables of interest in the script that which was written in Matlab. The processes used in the present study are common in running biomechanics analysis. There is nothing inherently unique about the use of Matlab to conduct these analyses. It is only the programming platform used to perform our post-processing steps.

Has the gait cycle analysis developed specifically for this research work?

Response:

  • No, the gait cycle analysis used is not unique to this study. In fact it is very common in the field of human locomotion biomechanics, including walking and running gait. The definition of gait cycle is commonly inclusive of the cycle from the initial contact of a single foot until the subsequent initial contact of the same foot. This cycle will always include a stance and swing phase. It is common to report variables for stance phase that are relevant to the dynamics of loading through the leg during this phase.

Why did you choose the midpoint to compute the position of the rearfoot? Are there any research works in literature to confirm your choice? The authors should better justify this choice.

Response:

  • The midpoint of the medial and lateral calcaneus markers was chosen to reflect subtalar joint motion (inclusive of calcaneal eversion in the frontal plane). Previous investigators have noted the subtalar joint center is approximately 12 mm below the tibio-talar joint center. Taking this into account, we used the medial and lateral calcaneus markers to estimate a central point in the subtalar joint, which was described as the rearfoot coordinate system. We assume the rearfoot moves as a rigid body, and therefore the whole rigid body will rotate as one system. The rigid body assumption for a skeletal element is very common in human locomotion biomechanics. This midpoint was used to represent a central point to calculate rotations and displacements from. The approach of using posterior heel marker, medial, and lateral calcaneus markers to define the rearfoot coordinate system is common to studies focused on rearfoot motion.

An image about the main workflow of the data analysis should be added to better clarify the use of Matlab.

Response:

The approaches described in the methods section are common to human locomotion biomechanics. We think that the clarification of the post-process analysis procedures referenced in the previous response above is sufficient to address the workflow of our data analysis approach, and that a figure showing the overall workflow would be distracting from the findings of the study.

Reviewer 2 Report

Increasing Step Rate Affects Rearfoot Kinematics and Ground Reaction Forces During Running

 BIOLOGY

 The purpose of this study was to evaluate the effects of increasing step rate on kinematics at the rearfoot in the sagittal, frontal, and transverse planes during the stance phase of running and to confirm how ground reaction forces throughout the stance phase are adjusted with increased step rate. The authors hypothesized there would be significant reductions throughout stance in rearfoot angles and ground reaction forces.

The manuscript reads smoothly and is easy to understand.  The aims, scope, and results of the study are clearly stated.  I have very much enjoyed reading this paper. I find it interesting and clearly written, and satisfying also all the other publication criteria of the “BIOLOGY JOURNAL”. The study provides a very valuable addition to this line of research, and adds relevantly to the subject with additional original findings. I thus find that this paper definitively delivers results that will surely be of interest to the readership of the “BIOLOGY JOURNAL”.  

  I recommend the publication of this interesting paper after minor revision:

  • Please add the number of the Research Committee,
  • Can we modelize the movement for the better prevention? Can you develop this point LIKE a perspective of this work?

I recommend to authors to increase the number of references with the related work, and thus, recommend the utilization of the following references for the discussion improvement:

*El-Ashker, S., Hassan, A., Taiar, R., Tilp, M., (2018). Long jump training emphasizing plyometric exercises is more effective than traditional long jump training: A randomized controlled trial, Journal of Human Sport and Exercise, 14(1), 215-224 doi:https://doi.org/10.14198/jhse.2019.141.18.

Author Response

We thank the reviewers for taking the time to review this manuscript, and for the positive feedback. We have revised the manuscript accordingly, and are confident that the manuscript has been improved through the review process. Please find point by point responses to comments below.

Reviewer 2

Please add the number of the Research Committee

Response:

  • We are unsure what the reviewer is requesting with this comment. The IRB protocol number is included at the end of the document.

Can we modelize the movement for the better prevention? Can you develop this point LIKE a perspective of this work?

Response:

  • Again, we are not sure what the reviewer is requesting with this comment. We would be happy to address further if clarification is provided.

I recommend to authors to increase the number of references with the related work, and thus, recommend the utilization of the following references for the discussion improvement: *El-Ashker, S., Hassan, A., Taiar, R., Tilp, M., (2018). Long jump training emphasizing plyometric exercises is more effective than traditional long jump training: A randomized controlled trial, Journal of Human Sport and Exercise, 14(1), 215-224 doi:https://doi.org/10.14198/jhse.2019.141.18.

Response:

  • We do not see the relevance of this work in relation to the present manuscript. The suggested reference appears to focus on a plyometric intervention for long jump training. We cannot find the overlap with our work, as we did not perform any type of training intervention over a period of time with a retest of performance, and were focused on running kinematics, not long jump performance.

Reviewer 3 Report

Summary

This study describes a sophisticated analysis of the changing kinematics  in the foot and foot's Ground Reaction Forces (GRF) when runners run at preferred speed,  5% faster than preferred speed and at 10% faster than preferred speed, the latter two conditions acoustically cued by means of a metronome. Hypotheses about reduced motion in the ankle at higher speeds and reduced GRFs are properly embedded in earlier research.  Twenty runners were investigated with state-of-the-art methods and analysis techniques among which Statistical Parameter Mapping to track at which phase of the stance interval of the right foot  the statistically significant motion changes occurred. Both the overall kinematic effects and the time-dependent effects during the stance phase confirmed the hypotheses. Their relevance for prevention of runner-related injuries is succinctly and convincingly discussed. 

General comments.

This is a well written, high-quality research report, with a focussed question, properly embedded in earlier research, sophisticated methodology and interesting results that are of interest to the readership of the journal. In the abstract of the paper, the main effects  of increased running speed on the kinematics and kinetics could be specified in somewhat more detail rather than referring only to 'changes', i.e. effects without indication of the direction of the effects (line 19 and line 32) but everything else reported in this paper is of high quality. I recommend publication after minor revision of the matter just mentioned. 

Detailed comments.

No further comments. 

Author Response

We thank the reviewers for taking the time to review this manuscript, and for the positive feedback. We have revised the manuscript accordingly, and are confident that the manuscript has been improved through the review process. Please find point by point responses to comments below.

Reviewer 3

In the abstract of the paper, the main effects  of increased running speed on the kinematics and kinetics could be specified in somewhat more detail rather than referring only to 'changes', i.e. effects without indication of the direction of the effects (line 19 and line 32) but everything else reported in this paper is of high quality. 

Response:

  • Thank you for the encouraging words, and recommendation. We have modified the last sentence of the abstract to refer to decreases in peak angles, as opposed to ‘altering’ angles.

Round 2

Reviewer 1 Report

Dear authors,

I red your responses and I concluded that only a small parts of the requested information have been added to improve the quality of the paper.

  • Suggested research works to be cited: the two proposed reserach works describe the most innovative MOCAP technologies and present  how to develop software modules for motion capture data analysis in medical fields. There are several other reserach works that you can cite. I proposed a couple of them.
  • The standard gait analysis module with MATLAB. If there are no specific software modules developed you should mention that the data analysis about the rearfoot has been on the traditional gait analysis.
  • Discussion and conclusion. You reported:
    • "Despite these findings, further research is needed to solidify the connections between changes in rearfoot motion, tibial rotation, and RRI development."
    • "Further research is needed to quantify the effects these changes in rearfoot and tibial motion have on soft tissue and bone loads."
      I wuod like to suggest a deeper description of your positve results otherwise the paper could seem only a preliminary study.

After the proposed major review the paper can be considered for the pubblication.

Author Response

We thank the reviewer and editor for taking the time to review our revised manuscript. We have revised the manuscript accordingly, and are confident that the manuscript has been improved through the review process. Please find point by point responses to comments below.

Reviewer 1

I red your responses and I concluded that only a small parts of the requested information have been added to improve the quality of the paper.

Suggested research works to be cited: the two proposed reserach works describe the most innovative MOCAP technologies and present  how to develop software modules for motion capture data analysis in medical fields. There are several other reserach works that you can cite. I proposed a couple of them.

The standard gait analysis module with MATLAB. If there are no specific software modules developed you should mention that the data analysis about the rearfoot has been on the traditional gait analysis.

Discussion and conclusion. You reported:

  • "Despite these findings, further research is needed to solidify the connections between changes in rearfoot motion, tibial rotation, and RRI development."
  • "Further research is needed to quantify the effects these changes in rearfoot and tibial motion have on soft tissue and bone loads."

I wuod like to suggest a deeper description of your positve results otherwise the paper could seem only a preliminary study.

After the proposed major review the paper can be considered for the pubblication.

Response:

The suggested references include alternatives to motion capture or development of modules to analyze motion capture data. We agree utilizing new alternatives to motion capture, especially those that can be used outside of the laboratory environment, is important. We have included a reference from the suggested works to suggest a possible avenue for future research.

With respect to the standard gait analysis suggestion, we have added a statement in the methods section to say that the approach used for data analysis was traditional gait analysis.

We agree this was a preliminary study to investigate the effects of increasing step rate on rearfoot kinematics. We think that a thorough description of our results has been provided, and that the present findings have been adequately placed within the context of previous work, and should not overplay the results with extended descriptions. We believe future research is needed, as stated, to better understand how changing rearfoot kinematics affects tissue loads and interactions with other body segments. As these points were not the purpose of the present study, and were not evaluated, we cannot speak more to how these factors may be affected by this step rate intervention.